# Crafting Contours: A Comprehensive Guide to Scrotal Reconstruction

**DOI:** 10.3390/life14020223

**Published:** 2024-02-04

**Authors:** Shota Suda, Kenji Hayashida

**Affiliations:** Division of Plastic and Reconstructive Surgery, Faculty of Medicine, Shimane University, 89-1 Enya-cho, Izumo 693-8501, Japan; s.suda@med.shimane-u.ac.jp

**Keywords:** scrotum, reconstruction, perineum, Fournier’s gangrene, hidradenitis suppurativa

## Abstract

This review delves into reconstructive methods for scrotal defects arising from conditions like Fournier’s gangrene, cancer, trauma, or hidradenitis suppurativa. The unique anatomy of the scrotum, vital for thermoregulation and spermatogenic function, necessitates reconstruction with thin and pliable tissue. When the scrotal defect area is less than half the scrotal surface area, scrotal advancement flap can be performed. However, for larger defects, some type of transplantation surgery is required. Various options are explored, including testicular transposition, tissue expanders, split-thickness skin grafts, local flaps, and free flaps, each with merits and demerits based on factors like tissue availability, defect size, and patient specifics. Also, physicians should consider how testicular transposition, despite its simplicity, often yields unsatisfactory outcomes and impairs spermatogenesis. This review underscores the individuality of aesthetic standards for scrotal reconstruction, urging surgeons to tailor techniques to patient needs, health, and defect size. Detailed preoperative counseling is crucial to inform patients about outcomes and limitations. Ongoing research focuses on advancing techniques, not only anatomically but also in enhancing post-reconstruction quality of life, emphasizing the commitment to continuous improvement in scrotal reconstruction.

## 1. Introduction

The causes of scrotal defects vary (Table 1), and to date, there is no consensus about the best method of scrotal reconstruction. Therefore, a guiding paper on scrotal reconstruction according to the type of disease and defect is needed. The closure of a scrotal defect with remnant scrotal tissue has an advantage of the testes being covered with native tissue, so the appearance is similar to the original scrotum. According to Chen et al. and Maurya et al., when the scrotal defect area is less than half the scrotal surface area, scrotal advancement flap can be performed [1,2]. However, for larger defects, some type of transplantation surgery is required. One of the most common causes of scrotal defect is Fournier’s gangrene, a rapidly progressing necrotizing fasciitis of the genital and perineal tissues [3]. Breakdown of immunity in the urogenital and anorectal areas leads to bacterial infection, causing Fournier’s gangrene. Commonly isolated bacteria include *Staphylococcus aureus*, *Streptococcus species*, *Escherichia coli*, *Klebsiella species*, and anaerobic bacteria such as *Bacteroides fragilis* [4]. The incidence and severity of Fournier’s gangrene are increased in individuals with an immunocompromised state; for example, patients with diabetes are more susceptible to developing the condition [5]. In the general US population, the incidence is 1.6 cases per 100,000 men, with a mortality rate of 7.5% [6]. Early diagnosis and treatment, including aggressive removal of necrotic tissue, are essential in managing Fournier’s gangrene [3].

Although infrequent, scrotal defects also occur after surgical resection of scrotal skin cancer. The incidence of extramammary Paget’s disease in Europe is 0.11 to 0.6 per 100,000 people annually [7,8]. An epidemiological survey in China suggests that it is even rarer among Asians [9]. Diseases such as scrotal cancer and extramammary Paget’s disease require sufficient excision of the lesion and its surrounding margins for radical treatment [10].

Scrotal defects also occur due to trauma. The scrotal skin is more vulnerable to traumatic damage than the penis and testes [11], and traumatic scrotal defects are more common in developing countries [2]. Currently, genital injuries are often caused by traffic accidents [12]. Degloving injuries to the scrotal skin frequently occur due to accidents involving vehicles and industrial machinery [13,14]. Another cause of scrotal defects is hidradenitis suppurativa, an inflammatory disease characterized by chronic abscesses, fistulas, and scars in the axillae, groin, inframammary folds, and perianal regions [15]. This condition causes pain, chronic discharge of foul-smelling secretions, and disfigurement. It often leads to depression, significantly reducing quality of life [16]. Its prevalence ranges from 0.05% to 4.10% [15], and recurrent abscess drainage surgeries are often necessary in the affected areas. Removal of the affected tissue may become necessary, and in cases where the scrotum is involved, scrotectomy may be required [17].

Loss of the scrotum has a significant impact on males. Physically, it can lead to decreased reproductive capability and hormonal imbalances [18], and changes in sensation and appearance in the genital area. Psychologically, it can lead to decreased self-esteem and sexual confidence, identity crises, grief, and depression [19]. Intact male genitalia is very important for a man’s self-esteem [20]; conversely, disfigured genitalia can lead to anxiety and concerns about relationships with sexual partners. Reconstructing the scrotum contributes to the recovery of sexual function and reproductive capability, improvement in quality of life, restoration of physical identity and self-respect, and normalization of self-perception in sexual relationships. Hence, effective scrotal reconstruction following defects is imperative.

This review focuses on the various surgical treatments for total scrotal skin loss. Since the scrotal skin is soft and pliable, primary closure is possible for defects of a certain size. However, when more than two-thirds of the scrotal skin is missing, other reconstructive methods must be used to encase the testes [21]. Although numerous reconstruction methods have been proposed, a gold standard has yet to be established [22]. In this review, we propose a scrotal reconstruction approach that not only restores the anatomical integrity, but also maintains the endocrine and spermatogenic functions of the testes. In conducting this literature review, we followed a comprehensive approach to ensure coverage of relevant studies. Studies were identified in the PubMed and Semantic Scholar databases up to December 2023. The different databases were searched using the following set of search Medical Subject Headings (MeSH) terms, which described ‘scrotum’, ‘reconstruction’, ‘scrotal reconstruction’, ‘skin graft’, ‘flap’, or a combination of these terms. In clinical practice, scrotal reconstruction may be performed several times a year. However, there is no consensus on the optimal scrotal reconstruction method or timing. This manuscript will potentially contribute to the creation of a recognized best practice in the field of scrotal reconstruction.

**Table 1 life-14-00223-t001:** Indications for scrotal reconstruction.

Category	Specific Causes	Description	References
Infectious diseases	Fournier’s gangrene	Rapidly progressing necrotizing fasciitis of the genital and perineal tissues, often due to bacterial infection in immunocompromised individuals, requiring early diagnosis and aggressive treatment	[3,4,5,6]
Tumors	Scrotal cancer, extramammary Paget’s disease	Need to excise lesion and surrounding skin for treatment; higher incidence noted in the European population	[7,8,9,10]
Traumatic conditions	Traffic accidents, industrial injuries	Common in developing countries, including degloving injuries from vehicles and machinery	[2,11,12,13,14]
Inflammatory disease	Hidradenitis suppurativa	Chronic disease characterized by abscesses, fistulas, and scars, mainly in the axillae, groin, inframammary folds, and perianal regions; requires surgical intervention in severe cases	[15,16,17,23]

## 2. Scrotal Anatomy and Functions

### 2.1. Scrotal Anatomy

The scrotum possesses unique structure and properties. It consists of an exceptionally thin skin and a smooth muscle layer called the dartos fascia [24]. The dartos fascia reacts to changes in external temperature by contracting or relaxing. In this regard, it works together with the cremaster muscle, which also moves the testes closer to the body for warmth or away from the body for cooling (Figure 1). Both the dartos fascia and the cremaster muscle help regulate the local temperature of the testes. This thermoregulation is one of the most crucial roles of the scrotum, and it ensures normal functioning of the testes [25].

### 2.2. Scrotal Functions

The testes are essential for spermatogenesis, and normal sperm production occurs only within a precise temperature range. Since sperm are sensitive to even slight temperature increases [26]—a rise of just 1.5 °C can significantly impair sperm production and lead to abnormalities [27]—it is crucial to keep the testes cooler than the body’s core temperature [28]. Therefore, in scrotal reconstruction, surgeons should select thin and pliable tissues that can help to maintain lower temperature [29]. In addition to spermatogenesis, another vital function of the testes is the production of testosterone. Approximately 95% of the testosterone in the blood is produced by the Leydig cells in the testes [30,31]. While a lower testicular temperature is necessary to maintain normal spermatogenesis, testosterone production by the Leydig cells does not decrease within the temperature range of 32–37 °C [32]. A previous report showed that levels of testosterone, luteinizing hormone, and follicle-stimulating hormone do not change after scrotal reconstruction, regardless of the method [33]. This implies that even at relatively high temperatures, testosterone production remains stable. Therefore, in scrotal reconstruction, the priority is to preserve spermatogenic function. 

## 3. Approaches to Scrotal Reconstruction

### 3.1. Testicular Transposition

Transposing the testes to the proximal thigh is a longstanding surgical procedure [34]. This involves creating a pocket by dissecting the subcutaneous tissue of the thigh, and moving the testes and spermatic cord into this space. This approach is adopted due to its simplicity. However, it has several disadvantages. First, inserting the testes under the thigh skin can cause pain and discomfort during walking or sitting [35]. Secondly, the aesthetic outcome is poor since the scrotum remains absent. Moreover, moving the testes to the proximal thigh impairs spermatogenesis by increasing their temperature. It is known that testes transposed to the groin or abdomen experience an increase in temperature, leading to a decrease in sperm production and abnormalities in sperm morphology [27,36,37]. Therefore, testicular transposition should be adopted only as a temporary means to protect the testes, and it will be necessary to reconstruct the scrotum later using a different approach.

### 3.2. Tissue Expander

The application of a tissue expander (TE) in scrotal reconstruction was first reported in 1990 [38]. The procedure involves temporarily inserting a TE under the skin to stretch the dermal layer, after which the testes and spermatic cord are placed beneath the expanded skin. In a case reported by Rapp et al., a TE was used when a fragment of normal-appearing scrotal skin measuring 3 cm by 6 cm remained [39]. A two-stage scrotal reconstruction was planned, using a TE with dimensions of 6.9 cm by 3.2 cm by 3.2 cm and a volume of 50 cm^3^. Over approximately 2 months, the TE was gradually expanded to about 150 cm^3^. Four months after inserting the TE, the testes and spermatic cord were placed inside the expanded scrotal skin. However, the reconstructed scrotum adhered to the perineum. This demonstrates the limitations in the size of scrotal skin that can be expanded with a TE. Using only stretched skin may not recreate the scrotum’s natural, hanging shape. Moreover, a TE is only applicable if some scrotal skin remains; it is not an option when there is complete scrotal loss. Additionally, it takes over 1 month for the TE to expand [39,40]. Surgeons must also fully inform patients of the discomfort that occurs during the TE insertion period, and patients must endure this condition. Thus, scrotal reconstruction using a TE imposes physical and psychological burdens on patients, and its range of application is limited.

### 3.3. Split-Thickness Skin Graft

Since the first report of its use in 1956 [41], split-thickness skin graft (STSG) has been widely adopted for scrotal reconstruction. The technique is technically simple [35,42] and can easily cover extensive skin defects. Therefore, it is often the preferred choice for reconstruction cases with total scrotal loss. However, the spermatic cord, which is the tubular structure that suspends the testes and epididymis in the scrotum from the abdominal cavity, tends to shrink and immobilize after inflammation. When performing surgery with a skin graft, it is recommended to suture the testes together via the tunica vaginalis and position them in a low position away from the perineum (Figure 2A,B).

STSG plays a critical role in rapidly restoring the skin’s barrier function and preventing infection. Meshed STSG is commonly preferred because it is more resistant to infection and tends to integrate better than sheet grafts [43]. Recently, to improve the rate of graft integration, perioperative negative pressure wound therapy has sometimes been used with STSG [44,45,46]. Negative-pressure wound therapy maintains an ideal wound healing environment and ensures better adhesion of the graft, thus enhancing both the survival rate and the morphological outcome of the reconstructed scrotum [44]. 

However, scrotums reconstructed with STSG present multiple problems. STSG inevitably leads to contracture due to postoperative shrinkage, and areas reconstructed with STSG are vulnerable to mechanical stress like friction or pressure [47]. Furthermore, the movement of the cremaster muscle is impeded, resulting in loss of the cremasteric reflex and its protective action on the testes during everyday activities. The reduced mobility of the reconstructed scrotum impairs temperature regulation, leading to an increase in testicular temperature. Consequently, and of utmost concern, spermatogenesis may decline after reconstruction by grafting [37,48]. Therefore, when considering STSG reconstruction for younger patients, they must be fully informed about the potential risks to fertility. Moreover, a scrotum reconstructed with an STSG often has suboptimal cosmetic appearance, resulting in patient dissatisfaction. In response to these limitations, there have been proposals for secondary cosmetic surgeries to enhance the appearance of scrotums reconstructed with STSG [49]. Reconstructive surgeons need to recognize that simply covering the testes with skin may diminish the patient’s quality of life and lead to psychological distress [50,51,52].

### 3.4. Local Flaps

Local flaps offer robust tissue resistance to mechanical stresses, such as friction. The elevation of local flaps is straightforward for most methods. The use of nearby perineal tissue allows for excellent color matching (replaces like tissue with like), and the donor site is easily concealed by underwear, thus not drawing attention postoperatively [53,54]. While skin grafts limit the function of the cremaster muscle, local flaps preserve this function, allowing the testes to move up and down in response to environmental temperature changes [35]. Also, flaps can be used for female-to-male sex reassignment surgery to cover the artificial testes. Local flaps, being less prone to shrinkage than split-thickness grafts, help prevent scrotal contracture. The absence of contracture means patients will not experience pain or discomfort while walking due to a tightened reconstructed scrotum. If a thin local flap is selected, it can release heat in the scrotum, preventing an increase in testicular temperature and thereby maintaining spermatogenic function [55,56]. Thus, reconstructing the scrotum with thin, supple, and durable local flaps can enhance the patient’s quality of life during daily activities and in terms of reproductive functions [53,54]. 

In current practice of scrotal reconstruction, bilateral local flaps are frequently employed due to their effectiveness in accurately recreating the scrotum’s natural morphology [57,58]: unilateral flaps do not provide enough tissue to achieve this (Figure 3). Using bilateral flaps provides enough tissue to prevent the testes from adhering to the perineum and recreate the natural hanging shape of the scrotum [59]. Furthermore, it allows for the formation of a scrotal septum [60], which can prevent testicular torsion. The choice between using a unilateral flap or bilateral flaps for reconstruction depends on various factors, including the size and location of the defect [61,62], the patient’s anatomy and health status, and the specific goals of the reconstruction. For large defects, bilateral flaps may provide more tissue for coverage and thus be more suitable. In contrast, smaller defects may be adequately covered with a unilateral flap. With respect to location, bilateral flaps may be needed if the defect extends over a large area that cannot be covered by a single flap. The choice may also depend on the patient’s anatomy and the availability of suitable donor sites. For instance, if one side has been debrided and incised, only a unilateral flap is feasible. Additionally, the patient’s overall health and ability to tolerate a longer, more complex surgery is a crucial factor in the case of bilateral flaps. Bilateral flap surgery is generally more complex and may result in a longer recovery time. This needs to be considered against the potential benefits. Furthermore, bilateral flaps may offer more symmetrical reconstruction but can also result in more scarring and potential morbidity at the donor sites. Therefore, the choice of unilateral or bilateral flaps should be based on a thorough individualized evaluation of the patient, including a discussion of the risks, benefits, and expected outcomes of each approach.

Options for local flaps in scrotal reconstruction are available in three areas (Table 2): the groin, medial thigh, and lateral thigh. In the groin, superficial circumflex iliac perforator (SCIP) flaps are used [63,64,65,66]. In the superomedial thigh region, gracilis flaps nourished by the medial circumflex femoral artery and perforator flaps of the anterior branch of the obturator artery are selected [67,68,69,70,71]. In the lateral thigh, anterolateral thigh (ALT) flaps based on the lateral circumflex femoral artery are used [72]. Among these, the most appropriate flap is selected based on the individual case.

The SCIP flap is the thinnest flap available in the perineal area, with an average thickness of 5–7 mm due to minimal subcutaneous fat [78,79]. Ando et al. demonstrated that spermatogenic function could be preserved using these thin flaps for scrotal reconstruction [55]. They elevated the SCIP flap in the superficial fascial layer and used bilateral super-thin SCIP flaps to wrap around the testes. They assessed the quantity and quality of sperm postoperatively and found that spermatogenic function improved over time. This suggests that thin flaps like the SCIP flap may help preserve spermatogenic function after scrotal reconstruction. Other local flaps in the perineal area are thicker than SCIP flaps. The ALT flap is typically raised as either a fasciocutaneous or a myocutaneous flap [73]. The thickness of the fasciocutaneous ALT flap in Asians averages about 9.8 mm [80], and it is known to correlate with body mass index [81]. In obese patients, the flap will be even thicker, and therefore less preferable to SCIP flaps [51]. Other perforator flaps from the medial thigh also tend to accumulate subcutaneous fat, making their thickness dependent on the patient’s degree of obesity [75]. 

Therefore, bilateral SCIP flaps should be the primary approach for scrotal reconstruction as local flaps, and other local flaps should only be chosen when SCIP flaps are not viable due to debridement. Hence, several papers described minor complications such as margin necrosis in a flap reconstruction group. However, direct closure was performed for wound dehiscence and infection was managed by frequent dressing [3,82]. Every flap used to cover scrotal defect has its own pros and cons, so every case needs an individual approach.

### 3.5. Free Flaps

Free flaps are also an effective but challenging option for scrotal reconstruction. The greatest advantage is the ability to select tissues that can appropriately replicate the characteristics of the scrotum regardless of their anatomical location. In cases like Fournier’s gangrene, which is characterized by an infection that extends widely across the perineal soft tissues, extensive debridement may be required, and it may not be possible to elevate a sufficiently large local flap. If the size of the elevated flap is small, it could result in a poorly reconstructed scrotum that appears adhered to the perineal area. Free flaps, in contrast, allow for the selection of tissues that can better replicate the scrotal properties without being limited to the perineal area. So far, there have been few ambitious reports of scrotal reconstruction using free flaps.

Yamakawa et al. reconstructed a scrotal defect following Fournier’s gangrene using an ulnar forearm free flap (UFFF) [50] (Figure 4). The size of the UFFF used for reconstruction was 22 × 10 cm^2^, and a full-thickness skin graft was performed at the donor site. The scrotum reconstructed with the UFFF hung down to the perineal area when the patient was in standing position, exhibiting an excellent shape. Ulnar forearm flaps are generally thin and pliable, and they are often used for oral reconstruction [83]. An ulnar forearm flap is less conspicuous than a radial forearm flap [84,85] and has a lower rate of donor site morbidity [83,86,87], making it a more favorable option for a free flap procedure. Although its vascular pedicle is about 4 cm shorter than that of the radial forearm flap, it is sufficiently long relative to other commonly used free flaps [87]. Since UFFF allows for harvesting of flaps larger than 20 cm, it enables a highly flexible reconstruction design. Additionally, a significant advantage of this technique is the capacity to harvest a thin flap regardless of the patient’s body type. This characteristic is particularly beneficial as it ensures uniformity in flap thickness across a diverse range of patient profiles. It thus enhances the versatility and adaptability of the procedure in various clinical situations. On the other hand, a disadvantage of the UFFF is the difficulty of vascular anastomosis due to the small diameters of the vessels [86,88]. Another drawback is the need for skin grafting at the donor site. When a large flap is required for scrotal reconstruction, primary closure at the donor site is not possible, thus necessitating skin grafting. Patients may not be satisfied with the grafted area on the arm; therefore, it is crucial to clearly inform them about the need for skin grafting on the ulnar forearm, confirm their understanding, and obtain their consent.

Another option for free flaps is the medial sural artery perforator (MSAP) free flap used by Teven et al. [52]. A 25 cm × 10 cm MSAP free flap was used to reconstruct the scrotum and perineal area, and it exhibited an excellent shape. The MSAP free flap is thin and flexible, and it is commonly used for head, neck, and limb reconstructions [89,90]. The vascular pedicle is relatively long, averaging about 10 cm [91]. The donor site is less conspicuous, and the rate of morbidity at this site is low [90]. The thickness of the MSAP free flap is about 5–12 mm [92], making it suitable for reconstructing a thin scrotum. However, there is a risk of not being able to harvest a sufficient flap due to vascular anomalies [91]. This makes it unsuitable for use in some cases. Therefore, careful evaluation of the vasculature (using echography or computed tomographic angiography) is necessary when considering this flap for reconstruction [93]. 

A unique free flap method is the combination of a free greater omental flap with an STSG [94,95]. The use of a pedicled omental flap for scrotal and perineal reconstruction was first reported in 1994 [96]. The pedicled omentum was harvested and placed in the perineal area through a subcutaneous tunnel, followed by an STSG. At that time, the scrotum only appeared to adhere to the perineal area. On the other hand, a free omentum does not require passage through a subcutaneous tunnel. This allows for the use of a larger piece of omentum that is more easily processed into a bag shape for a reconstructed scrotum. An STSG adheres easily on the omentum, and once adhered it is less prone to contraction, and thus more durable. However, a laparoscopic procedure is necessary to harvest this flap, and this necessitates the support of a gastrointestinal surgeon. Furthermore, since the surgery involves multiple stages, it is complex and requires a longer operation time. Additionally, it leads to a risk of intra-abdominal infection and postoperative abdominal wall hernia [97]. These factors complicate postoperative management and increase the burden on the patient. Consequently, this approach is considered less favorable.

It can be said that for all free flap methods, postoperative leg movements stress the anastomosis site. This can lead to flap thrombosis or congestive necrosis. However, despite these risks, free flaps are an excellent method for addressing extensive scrotal defects and should be considered as a reconstruction option.

## 4. Conclusions

This review of various scrotal reconstruction techniques revealed a diverse landscape of surgical options, each with its unique advantages and limitations. The discourse on reconstruction methods not only focused on replicating the scrotal structure but also included reports on post-reconstruction testicular function, specifically spermatogenic function and testosterone production [18,29,37,55]. According to the studies reviewed, reconstructing the scrotum with thin flaps leads to preserved spermatogenic function, while reconstruction with thick flaps does not. Thus, thin flaps that do not increase testicular temperature are essential for fertility-preserving scrotal reconstruction. 

Considering the basic principle of reconstruction, namely ‘replaces like tissue with like’, using thin bilateral SCIP flaps from the local tissue near the scrotum is deemed most appropriate. However, there are cases in which this flap cannot be used due to the extent of debridement. In such cases, as a second-line approach, reconstruction using UFFF or MSAP free flaps should be considered.

Regarding aesthetics, it is impossible to universally define a ‘beautiful scrotum’ due to individual values [98]. In addition, once the wounds have healed, both physicians and patients may not care about the condition of the reconstructed scrotum, especially in elderly patients. However, the choice of reconstruction technique should be tailored to the individual patient, considering the extent of the defect, the patient’s overall health, and their specific needs and preferences. It is imperative for surgeons to engage in thorough preoperative counseling, discussing the potential outcomes and limitations of each method for the natural position of testes. This approach ensures that patients are well informed and can make decisions that align with their expectations and life circumstances.

Scrotal reconstruction remains a challenging yet evolving field. Continued research and innovation are necessary to refine the extant techniques and develop new methods that can better address the complex needs of patients undergoing scrotal reconstruction. The ultimate goal is to not only restore anatomy but also enhance the patient’s quality of life post reconstruction.

## Figures and Tables

**Figure 1 life-14-00223-f001:**
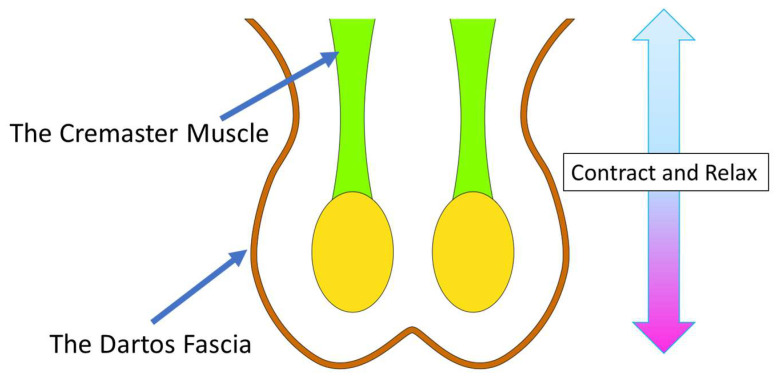
Functions of the scrotum. The dartos fascia collaborates with the cremaster muscle to contribute to testicular temperature regulation. The dartos fascia, a muscular layer, responds to temperature changes by contracting or relaxing. In synergy with the cremaster muscle, it adjusts testicular position for optimal temperature control.

**Figure 2 life-14-00223-f002:**
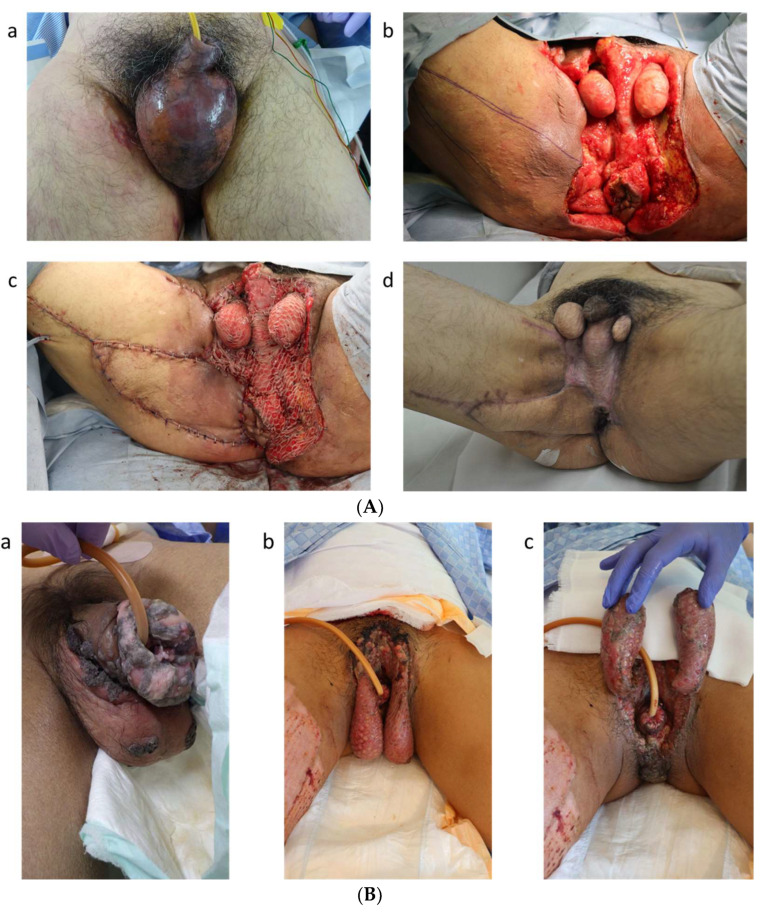
(**Aa**) A 60-year-old male presented with Fournier’s gangrene. (**Ab**) We performed debridement. (**Ac**) The right perineum was reconstructed with a gracilis muscle flap, and the scrotum was treated with meshed STSG. Both testes were adhered to the perineum by granulation tissue and closely attached to the perineal area. (**Ad**) Five months after scrotal reconstruction surgery. The scrotum was adherent to the perineal area and completely immobile. (**Ba**) A 43-year-old male. He suffered from malignant tumors of the penis and scrotum. Total penectomy and scrotectomy were performed. (**Bb**,**Bc**) The testes were covered with meshed STSG. After reconstruction, the testes are separated from the perineal area, but the movement of the cremaster muscle is restricted. Due to the lack of a pouch-like scrotum, the aesthetic appearance is inferior.

**Figure 3 life-14-00223-f003:**
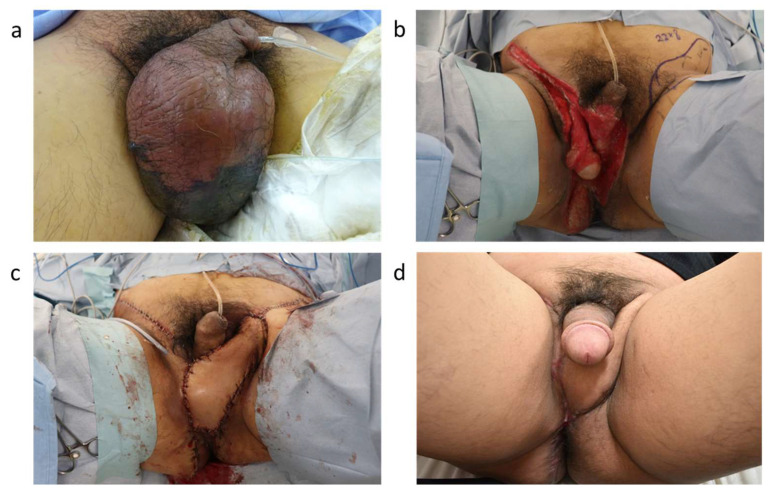
(**a**) A 43-year-old male presented with Fournier‘s gangrene. (**b**) After debridement. The right inguinal area was incised down to the muscle layer. The left testis was excised. (**c**) A unilateral SCIP flap was designed for the left inguinal area. Due to the use of a unilateral flap, there was insufficient tissue to recreate the form of the scrotum, and the testis adhered closely to the perineal area. (**d**) 6 months after the reconstruction. The reconstructed scrotum is slightly bulky.

**Figure 4 life-14-00223-f004:**
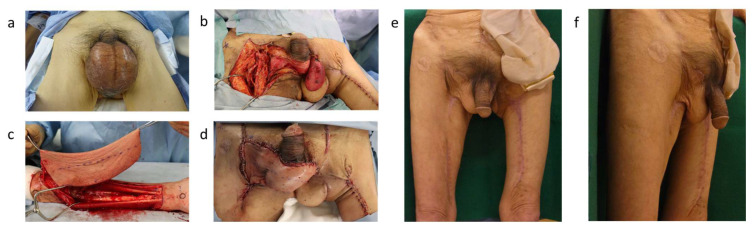
(**a**) A 74-year-old male presented with Fournier’s gangrene. (**b**) After debridement, both testes were exposed. The left pelvic floor was reconstructed with a left gracilis muscle flap. (**c**) The UFFF was raised from the left arm. We selected the profunda femoris artery and veins as the vascular anastomosis for the transplant bed. (**d**) The scrotum was reconstructed with UFFF, covering both testes. (**e**,**f**) Fourteen months post operation, the scrotum hangs down and has an aesthetically pleasing appearance. The mobility of the cremaster muscle is preserved.

**Table 2 life-14-00223-t002:** Three representative flaps commonly used for scrotal reconstruction.

	Advantages	Disadvantages	Unilateral Side	Bilateral Sides
SCIP flap (groin flap)	Thinnest flap in the perineal area (5–7 mm); minimal subcutaneous fat, making it suitable for preserving spermatogenic function. Postoperative improvement in spermatogenic function has been demonstrated. Ideal for bilateral use in scrotal reconstruction.	Limited tissue availability; may not be sufficient in cases of extensive debridement. Its thinness may be a limitation in patients requiring more voluminous tissue coverage.	[63,64,65,66]	[33,55,60]
ALT flap	Can be raised as either a fasciocutaneous or a myocutaneous flap; long pedicle with extensive reach; allows tailored reconstruction based on defect requirements; generally well-tolerated tissue transfer; low morbidity at the donor site.	Average thickness is about 9.8 mm in Asians, which is thicker than SCIP flaps. Thickness correlates with BMI, potentially limiting its use in obese patients. In cases where thin flaps are preferred for preserving spermatogenic function, ALT flaps may not be the ideal choice.	[51,56,72,73]	[57]
Medial thigh flap	Good option when SCIP flap is not viable; offers adequate tissue for reconstruction; can provide color and texture match similar to native scrotal tissue; if the muscular component is attached, useful for filling the dead space and for separating the anus from the perineum.	Tends to accumulate subcutaneous fat, making thickness dependent on obesity. The relative thickness compared to SCIP flaps may compromise the preservation of spermatogenic function.	[33,61,67]	[33,58,59,62,68,69,70,74,75,76,77]

These flaps can be used on a unilateral side or elevated from bilateral sides. BMI, body mass index; SCIP, superficial circumflex iliac perforator; ALT, anterolateral thigh.

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
