# Peer review of "Crafting Contours: A Comprehensive Guide to Scrotal Reconstruction"

_life, 2024, doi:10.3390/life14020223_

Round 1
Reviewer 1 Report
Comments and Suggestions for Authors
This is a nice comprehensive paper with nice figures. I would like more of a management theme for hydradinitis and lymphoedema- probably the most common cause of scrotal reconstruction. With hydradinitis there are also multiple stages until all the infection has cleared. In lymphoedema the lateral scrotal folds are often not involved and can therefore be used for reconstruction.
What about foreign bodies- silicone?
Also I agree with never placing the testes in the thigh- they should be left in the scrotal area and kept moist in a dressing until ready for reconstruction. Also the testes should be sutured together via the tunica vaginalis and sutured to a natural position- often as the recon is delayed granulation tissue will form over then enabling an easier ssg take and better cosmesis.
Inclusion of the above will improve the article.
Author Response
Reviewer 1
This is a nice comprehensive paper with nice figures. I would like more of a management theme for hydradinitis and lymphoedema- probably the most common cause of scrotal reconstruction. With hydradinitis there are also multiple stages until all the infection has cleared. In lymphoedema the lateral scrotal folds are often not involved and can therefore be used for reconstruction.
Thank you for your valuable comments. As you pointed out, severe HS needs surgical reconstruction. We could find a study published in JAMA Surgery suggests that genital HS requires medical and surgical management as well as close collaboration among a multidisciplinary team especially in genital area. So, we added the paper in Table.
In lymphedema cases, as you mentioned, the lateral scrotal folds can be used for reconstruction. So, we added some references related to lymphedema in local flap section.
(53) Ehrl, D.; Heidekrueger, P. I.; Giunta, R. E.; Wachtel, N. Giant Penoscrotal Lymphedema-What to Do? Presentation of a Curative Treatment Algorithm. J Clin Med 2023, 12 (24). DOI: 10.3390/jcm12247586 From NLM PubMed-not-MEDLINE.
(54) Kumar, S.; Saha, A.; Kumar, S.; Singh, P.; Singh, K. K. Giant Scrotal Lymphoedema: A Case Series. Cureus 2023, 15 (11), e48248. DOI: 10.7759/cureus.48248 From NLM PubMed-not-MEDLINE.
What about foreign bodies- silicone?
Foreign bodies including silicone are mainly used for female-to-male sex reassignment surgery. As you know, in that case, physicians should cover it using flaps. So, we add the sentences in local flap section.
Also I agree with never placing the testes in the thigh- they should be left in the scrotal area and kept moist in a dressing until ready for reconstruction. Also the testes should be sutured together via the tunica vaginalis and sutured to a natural position- often as the recon is delayed granulation tissue will form over then enabling an easier ssg take and better cosmesis.
Inclusion of the above will improve the article.
Thank you for your valuable comments. I totally agree with your suggestion to prevent testes adhered to thigh. We add these sentences in skin graft section.
“However, the spermatic cord, which is the tubular structure that suspends the testes and epididymis in the scrotum from the abdominal cavity, tends to shrink and immo-bilize after inflammation. When performing surgery with a skin graft, it is recom-mended to suture testes together via the tunica vaginalis and position them in a low position away from the perineum”.
Reviewer 2 Report
Comments and Suggestions for Authors
In Abstract,
· The aim of the review is confusing and should be clarified: did they review all type of scrotal reconstruction? Partial, total or both scrotal defects? In case of partial scrotal defects, the reconstructive technique is incomplete and should be in-depth. Therefore, the same insights have to be described and discussed in the main text.
In the Main text:
In the “Methods”
· Methodology is poor throughout the text. Authors have to describe how they identified the topic that were reviewed and how they conducted the literature search. Moreover, more information about inclusion or exclusion criteria should be reported.
In the “Approaches to Scrotal Reconstruction”
· The most challenging defects for plastic surgeons are the perineal ones, due to the natural predisposition of this area to maceration, infection, and unfavourable position of the patient during bed rest. In case of STSG reconstruction, its delicate process of engraftment, the poor aesthetic and functional result, and the challenges to healing in this area including a high bacterial bioburden, an inherently moist surgical site and forces which threaten to disrupt the site as the patient begins to mobilize severely limit the application. This aspect should be clarified.
· The introduction of thin and ultrathin flaps has changed the way to reconstruct this kind of defect. The versatility of SCIP flap in perineal reconstruction and its possible role in lymph drainage is well documented in Literature and deserves to be mentioned. Please cite relevant reference (PMID: 34866009, 35205824).
· Advancement V-Y thigh flap and VRAM flap should be described as possible reconstructive option in case of total scrotal defect.
Author Response
Reviewer2
In Abstract,
- The aim of the review is confusing and should be clarified: did they review all type of scrotal reconstruction? Partial, total or both scrotal defects? In case of partial scrotal defects, the reconstructive technique is incomplete and should be in-depth. Therefore, the same insights have to be described and discussed in the main text.
Thank you for your valuable comments.
According to Chen et al. and Maurya et al., when the scrotal defect area is less than half the scrotal surface area, scrotal advancement flap can be performed1), 2). However, for larger defects, some type of transplantation surgery is required.
(1) Chen, S. Y.; Fu, J. P.; Wang, C. H.; Lee, T. P.; Chen, S. G. Fournier gangrene: a review of 41 patients and strategies for reconstruction. Annals of plastic surgery 2010, 64 (6), 765-769. DOI: 10.1097/SAP.0b013e3181ba5485 From NLM Medline.
(2) Maurya, R.; Mir, M. A.; Mahajan, S. Various Options for Scrotal Reconstruction: A Prospective Observational Study. Cureus 2022, 14 (2), e22671. DOI: 10.7759/cureus.22671 From NLM.
So, we added these sentences in Introduction and Abstract.
In the Main text:
In the “Methods”
- Methodology is poor throughout the text. Authors have to describe how they identified the topic that were reviewed and how they conducted the literature search. Moreover, more information about inclusion or exclusion criteria should be reported.
We totally agree with you. This is a narrative review. When we conduct a literature review, it is crucial for them to provide transparency regarding their methodology for selecting relevant literature.
So, we added these sentences as follows:
In conducting this literature review, we followed a comprehensive approach to ensure coverage of relevant studies. Studies were identified in the PubMed and Semantic Scholar database up to December 2023. The different databases were all searched using the following set of search Medical Subject Headings (MeSH) terms which described ‘scrotum’, ‘reconstruction’, ‘scrotal reconstruction’, ‘skin graft’, ‘flap’ or a combination of these terms.
In the “Approaches to Scrotal Reconstruction”
- The most challenging defects for plastic surgeons are the perineal ones, due to the natural predisposition of this area to maceration, infection, and unfavourable position of the patient during bed rest. In case of STSG reconstruction, its delicate process of engraftment, the poor aesthetic and functional result, and the challenges to healing in this area including a high bacterial bioburden, an inherently moist surgical site and forces which threaten to disrupt the site as the patient begins to mobilize severely limit the application. This aspect should be clarified.
Split thickness skin grafting is a safe, cost-effective method of scrotal defect reconstruction that is technically easy to perform. Also, it can be done in short operative time to cover difficult defects. Hence, several papers described minor complications such as margin necrosis in a flap reconstruction group. (Ferreira PC, et al. Plast Reconstr Surg. 2007, Spyropulou GA, et al. Urology. 2013). However, direct closure was done for wound dehiscence and infection was managed by frequent dressing(Ferreira PC, et al. Plast Reconstr Surg. 2007).
So, each case of scrotal defect needs an individualized approach for tailoring the scrotal reconstruction depending upon patient’s age, general condition of patient, wound status, comorbidities, patient’s requirement and the advantages and disadvantages of the different management modalities will definitely help in decision-making for soft tissue reconstruction of the scrotum under optimal conditions.
We added “Hence, several papers described minor complications such as margin necrosis in a flap reconstruction group. However, direct closure was done for wound dehiscence and in-fection was managed by frequent dressing. Every flap used to cover scrotal defect has its own pros and cons, so every case needs an individual approach” in flap section.
- The introduction of thin and ultrathin flaps has changed the way to reconstruct this kind of defect. The versatility of SCIP flap in perineal reconstruction and its possible role in lymph drainage is well documented in Literature and deserves to be mentioned. Please cite relevant reference (PMID: 34866009, 35205824).
Thank you for your suggestion. Although 34866009 and 35205824 described vulvar reconstruction surgery, these references about SCIP flap are related to scrotal reconstruction. So we added them in flap section.
- Advancement V-Y thigh flap and VRAM flap should be described as possible reconstructive option in case of total scrotal defect.
In this literature search, VRAM flap was not used for scrotal reconstruction, but perineal reconstruction. However, VY thigh flap was generally used as we described Table 2. So, we added “if the muscular component is attached, useful for filling the dead space and for separating the anus from the perineum” in Table 2 and reference 78.
78) Thiele, J. R.; Weber, J.; Neeff, H. P.; Manegold, P.; Fichtner-Feigl, S.; Stark, G. B.; Eisenhardt, S. U. Reconstruction of Perineal Defects: A Comparison of the Myocutaneous Gracilis and the Gluteal Fold Flap in Interdisciplinary Anorectal Tumor Resection. Front Oncol 2020, 10.
Reviewer 3 Report
Comments and Suggestions for Authors
This is a very interesting review.
Please state how you selected the papers included in this review. Apparently it is a narrative review.
Are the pictures from authors' own work?
It would be interesting to know what is the timing between the different stages of reconstructions, according to the authors' experience.
How long patients can wait after surgical debriment, for reconstruction?
I have noticed the the aesthetic results of scrotal reconstruction is very pleasent after skin graft (whenever possible). What do authors think about the aesthetic point of view?
What is the revision rate at follow-up?
What their surgical site complication rate?
Author Response
Reviewer3
This is a very interesting review.
Please state how you selected the papers included in this review. Apparently it is a narrative review.
Thank you for your suggestion. As you pointed out, this is a narrative review. So, we added these sentences in Introduction.
In conducting this literature review, we followed a comprehensive approach to ensure coverage of relevant studies. Studies were identified in the PubMed and Semantic Scholar database up to December 2023. The different databases were all searched using the following set of search Medical Subject Headings (MeSH) terms which described ‘scrotum’, ‘reconstruction’, ‘scrotal recosntruction’, ‘skin graft’, ‘flap’ or a combination of these terms.
Are the pictures from authors' own work?
These photographs are own practice. So, we added this sentence “All case photos in this paper are from our own experience” in informed consent statement.
It would be interesting to know what is the timing between the different stages of reconstructions, according to the authors' experience.
Thank you for your suggestion. However, our case is not so much, but in clinical practice, scrotal reconstruction may be performed several times a year. Also, there is no consensus on the optimal timing of reconstruction.
So, we add “In clinical practice, scrotal reconstruction may be performed several times a year. However, there is no consensus on the optimal scrotal reconstruction method and timing .”in Introduction.
How long patients can wait after surgical debriment, for reconstruction?
As we mentioned above, the optimal timing of reconstruction is unknown. However, testes should contract within 2 or 3 weeks after initial debridement. So, we add these sentences. “However, the spermatic cord, which is the tubular structure that suspends the testes and epididymis in the scrotum from the abdominal cavity, tends to shrink and immobilize after inflammation. When performing surgery with a skin graft, it is recommended to suture testes together via the tunica vaginalis and position them in a low position away from the perineum“ in skin graft section.
I have noticed the the aesthetic results of scrotal reconstruction is very pleasent after skin graft (whenever possible). What do authors think about the aesthetic point of view?
We cannot agree with it. Maybe, if secondary reconstruction including skin graft is performed within 2 weeks, an aesthetic result would be acceptable. However, testes tend to adhere to the upper site of perineum. That depends on the timing of reconstruction or infection of the region.
What is the revision rate at follow-up?
In this literature review, we cannot find the paper which revealed the revision rate. Maybe, once the wounds have healed, both physicians and patients do not care about it. We add it in Conclusion.
What their surgical site complication rate?
Complications including surgical site infection are low based on these literatures. When physicians select reconstructive options, each physician should choose the reconstruction method according to their competence. We added “It is imperative for surgeons to engage in thorough preoperative counseling, discussing the potential outcomes and limitations of each method for the natural position of testes”.in Conclusion.
Reviewer 4 Report
Comments and Suggestions for Authors
Dear Editor and Authors,
Thank you for the opportunity to review the manuscript entitled “Crafting Contours: A Comprehensive Guide to Scrotal Reconstruction”. The authors aimed to review different aspects of aesthetic standards for scrotal reconstruction and highlighted the need to tailor techniques to patient needs, health, and defect size. The topic is very important, especially for surgeons (plastic surgeons, oncologic and general surgeons) and not so common in the literature However, I have some remarks:
- The authors should clearly state how they chose literature for their review – what aspects did they plan to analyze and which databases they searched for these issues.
- Are the photographs /cases/ presented are from authors’ own practice? If yes, congratulations on great results! And this should be clearly stated that these are your results! Having such experience in these defects, you could mention the number of cases of such problem you face annually to justify the need for such reviews.
- Also , consider mentioning the bilateral technique reported in: Thiele JR, Weber J, Neeff HP, Manegold P, Fichtner-Feigl S, Stark GB, Eisenhardt SU. Reconstruction of Perineal Defects: A Comparison of the Myocutaneous Gracilis and the Gluteal Fold Flap in Interdisciplinary Anorectal Tumor Resection. Front Oncol. 2020 May 6;10:668. doi: 10.3389/fonc.2020.00668.
Author Response
Reviewer 2
Thank you for the opportunity to review the manuscript entitled “Crafting Contours: A Comprehensive Guide to Scrotal Reconstruction”. The authors aimed to review different aspects of aesthetic standards for scrotal reconstruction and highlighted the need to tailor techniques to patient needs, health, and defect size. The topic is very important, especially for surgeons (plastic surgeons, oncologic and general surgeons) and not so common in the literature However, I have some remarks:
- The authors should clearly state how they chose literature for their review – what aspects did they plan to analyze and which databases they searched for these issues.
Certainly! When we conduct a literature review, it is crucial for them to provide transparency regarding their methodology for selecting relevant literature.
So, we added these sentences as follows:
In conducting this literature review, we followed a comprehensive approach to ensure coverage of relevant studies. Studies were identified in the PubMed and Semantic Scholar database up to December 2023. The different databases were all searched using the following set of search Medical Subject Headings (MeSH) terms which described ‘scrotum’, ‘reconstruction’, ‘scrotal recosntruction’, ‘skin graft’, ‘flap’ or a combination of these terms.
- Are the photographs /cases/ presented are from authors’ own practice? If yes, congratulations on great results! And this should be clearly stated that these are your results! Having such experience in these defects, you could mention the number of cases of such problem you face annually to justify the need for such reviews.
Thank you for your comments. These photographs are own practice. However, our cases is not so much. So, we added these sentences“All case photos in this paper are from our own experience” in informed consent statement. And “In clinical practice, scrotal reconstruction may be performed several times a year. However, there is no consensus on the optimal scrotal reconstruction method” in Introduction.
- Also , consider mentioning the bilateral technique reported in: Thiele JR, Weber J, Neeff HP, Manegold P, Fichtner-Feigl S, Stark GB, Eisenhardt SU. Reconstruction of Perineal Defects: A Comparison of the Myocutaneous Gracilis and the Gluteal Fold Flap in Interdisciplinary Anorectal Tumor Resection. Front Oncol. 2020 May 6;10:668. doi: 10.3389/fonc.2020.00668.
Thank you for your suggestion. This manuscript described the myocutaneous gracilis flap (MGF) and the gluteal fold flap (GFF) for defect coverage in extensive perineal/pelvic area.
As MGF should be included in Medial thigh flap, we added this in Table 2. Thank you.
Reviewer 5 Report
Comments and Suggestions for Authors
The authors present a review of the options for scrotal reconstruction. They provide thorough background of the condition and discuss the range of reconstructive options. They provide great references for each option and explain the pros and cons succinctly. They also provide figures from their own cases to add to the existing literature. My only comment is that the image quality for Figure 4, especially parts a-d is poor. If there is any way to have higher resolution images here, it would be ideal. Otherwise, the article is well written, and it will serve as a road map for surgeons in this space.
Author Response
Reviewer 3
The authors present a review of the options for scrotal reconstruction. They provide thorough background of the condition and discuss the range of reconstructive options. They provide great references for each option and explain the pros and cons succinctly. They also provide figures from their own cases to add to the existing literature. My only comment is that the image quality for Figure 4, especially parts a-d is poor. If there is any way to have higher resolution images here, it would be ideal. Otherwise, the article is well written, and it will serve as a road map for surgeons in this space.
Thank you for your words of praise we are very honored and did not expect such praise from reviewer. As you mentioned, this manuscript described the pros and cons for scrotal reconstruction methods sufficiently. Also, the image quality of Figure 4 is very important. But, unfortunately, these are the maximum resolution in Figure 4, but it should be acceptable. Thank you.
Reviewer 6 Report
Comments and Suggestions for Authors
The authors present a non-systematic review of scrotum reconstruction techniques.
The manuscript is certainly well written and provides interesting insights on a very specific topic but of considerable interest to urologists and plastic surgeons.
I have a few remarks to make:
Line 173-174: another limitation of STSG reconstruction to mention is the difficulty of performing secondary cosmetic surgery due to the adhesion of thin anatomic structures (spermatic funicle) to the graft, which may prevent safe dissection.
Line 187-188: kindly specify the type of thin flap you're referring to.
Line 240-241: an advantage of medial thigh-based flaps is that they allow for the filling of deep empty spaces while reconstructing the skin layer. Additionally, this solution can provide enhanced protection and separation between the anus and the scrotal wound by utilizing its muscular component.
Author Response
Reviewer 4
The authors present a non-systematic review of scrotum reconstruction techniques.
The manuscript is certainly well written and provides interesting insights on a very specific topic but of considerable interest to urologists and plastic surgeons.
I have a few remarks to make:
Line 173-174: another limitation of STSG reconstruction to mention is the difficulty of performing secondary cosmetic surgery due to the adhesion of thin anatomic structures (spermatic funicle) to the graft, which may prevent safe dissection.
We totally agree with you in this issue. The spermatic cord, which is the tubular structure that suspends the testes and epididymis in the scrotum from the abdominal cavity, is problematic for secondary reconstruction. So, we added these sentences in STSG section. “However, the spermatic cord, which is the tubular structure that suspends the testes and epididymis in the scrotum from the abdominal cavity, tends to shrink and immobilize after inflammation, ”
Line 187-188: kindly specify the type of thin flap you're referring to.
We added some recent references as you pointed out.
(49) Yamakawa, S.; Hayashida, K. Scrotal Reconstruction Using a Free Ulnar Forearm Flap: A Case Report. Journal of Plastic and Reconstructive Surgery 2022, 1 (1), 26-30. DOI: 10.53045/jprs.2021-0011.
(50) Sapino, G.; Gonvers, S.; Cherubino, M.; di Summa, P. G. The "Sombrero-Shape" Super-Thin Pedicled ALT Flap for Complete Scrotal Reconstruction Following Fournier's Gangrene. Arch Plast Surg 2022, 49 (3), 453-456. DOI: 10.1055/s-0042-1748663 From NLM.
(51) Teven, C. M.; Yu, J. W.; Zhao, L. C.; Levine, J. P. Extended medial sural artery perforator free flap for groin and scrotal reconstruction. Arch Plast Surg 2020, 47 (4), 354-359. DOI: 10.5999/aps.2019.01921 From NLM PubMed-not-MEDLINE.
Line 240-241: an advantage of medial thigh-based flaps is that they allow for the filling of deep empty spaces while reconstructing the skin layer. Additionally, this solution can provide enhanced protection and separation between the anus and the scrotal wound by utilizing its muscular component.
We totally agree with this point. We added your valuable comments to Advantages of medial thigh flaps in Table2.
Round 2
Reviewer 1 Report
Comments and Suggestions for Authors
all fine
Reviewer 2 Report
Comments and Suggestions for Authors
The revisions suggested, taken correctly into account by the authors, improved the quality of the article, and I think that this work deserves to be published in the present form
Comments on the Quality of English Language
The manuscript is well written.